# Impact of Training Protocols on Lifting Velocity Recovery in Resistance Trained Males and Females

**DOI:** 10.3390/sports9110157

**Published:** 2021-11-19

**Authors:** Christian Houmann Amdi, Daniel John Cleather, Jamie Tallent

**Affiliations:** 1Faculty of Sport, Applied Health and Performance Sciences, St. Mary’s University, Twickenham, London TW1 4SX, UK; daniel.cleather@stmarys.ac.uk; 2School of Sport, Exercise and Rehabilitation Sciences, University of Essex, Colchester CO4 3SQ, UK; jamie.tallent@essex.ac.uk; 3Department of Physiotherapy, School of Primary and Allied Health Care, Faculty of Medicine, Nursing and Health Science, Monash University, Melbourne, VIC 3199, Australia

**Keywords:** sex differences, velocity, squat, fatigue, fatigability, 1RM, strength training

## Abstract

It has been suggested that sex differences exist in recovery following strength training. This study aimed to investigate the differences in recovery kinetics between resistance trained males and females following two different back squat (BSq) protocols. The first protocol (eight females and eight males) consisted of five sets of five repetitions at 80% of their one-repetition maximum (1RM) in the BSq (SMRT), while the second (seven females and eight males) consisted of five sets to muscular failure (MF) with a 4–6RM load (RMRT). The recovery was quantified with the mean concentric velocity (MV) at 80% of the 1RM immediately before and 5 min, 24, 48, and 72 h after the training protocol. Following the SMRT, a significant between-sex difference, favoring the females, was observed at 5 min, 24 h, and 48 h following the SMRT (*p* < 0.05, Effect Size (ES) = 1.01–2.25). Following the RMRT, only the males experienced a significant drop in performance after 5 min compared to the baseline (*p* = 0.025, ES = 1.34). However, no sex differences were observed at any timepoint (*p* > 0.05). These results suggest that males experienced more fatigue than females following a protocol where the volume relative to the 1RM was matched, while no differences in fatigue were evident following a protocol in which multiple sets were performed to MF.

## 1. Introduction

Following a bout of strenuous resistance training (RT), a temporary decrease in work capacity and neuromuscular function, termed “fatigue”, is observed [1,2]. This performance decrement is considered to be a result of both peripheral and central mechanisms [1,2]. Given sufficient recovery, the body will adapt, surpassing its previous neuromuscular capabilities [1,3]. Therefore, understanding the temporal characteristics of RT-induced fatigue will help practitioners optimize training prescription.

In trained humans, some studies have suggested that women may experience less muscle damage following strenuous resistance exercise when measuring serum markers [4,5,6], while others observe no sex differences [3,7]. Furthermore, women have been shown to be less fatigable and able to perform more repetitions at a given intensity than men [8,9,10]. This is possibly due to a larger proportion of type I muscle fibers, higher capillarization of the muscle tissue, increased blood flow, less mechanical arterial compression, and decreased dependency on glycogen during exercise [9,11,12]. Therefore, similar training programs might lead to different training responses between men and women.

In contrast, when looking at the force production loss after strenuous exercise, the research in trained humans is less clear [3,13,14,15,16]. Davies et al. (2018) observed a reduced fatigability in men and quicker long-term recovery, while Häkkinen (1994) found that women were less fatigued immediately following RT. Despite Häkkinen (1994) and Judge and Burke (2010) suggesting a female superiority for fatigue resistance and recovery, many studies have failed to detect between-sex differences in recovery kinetics following exercise in trained participants [13,14,15,16]. However, due to differences in the participant training experience, exercise selection, and training protocols, it is difficult to conclude whether the observed differences are a result of confounding variables, such as training methodologies or participant demographics. For example, in Davies et al. (2018) and Marshall et al. (2020), who observed a male recovery advantage and no sex differences, respectively, the male and female strength, relative to the fat free mass (FFM) or body mass (BM), were within 7% of each other. In contrast, Häkkinen (1993,1994) and Judge and Burke (2010), who suggested a female fatigability and/or recovery advantage, compared men with women who were 54–69% weaker relative to the FFM or BM. Furthermore, widely different training protocols have been used concerning exercise selection (back squat (BSq), bench press, or a full body session), number of hard sets (one to twenty sets), and training intensity (70 to 100% of one repetition maximum (RM)). Moreover, all the studies required the participants to perform at least one set to muscular failure (MF), with three studies performing multiple sets to MF [13,14,16]. While multiple sets to MF ensures a substantial amount of fatigue, which might be of interest for researchers, it may not be applicable to the real-world training practices of trained individuals. Consequently, there is a need for further investigation of how training status and training protocol impact the sexes’ response to RT, particularly when participants are not performing sets to MF.

Therefore, the aim of this study was to investigate sex differences following a relative 1RM volume matched training session without MF and a protocol in which the proximity to MF is matched, with both groups reaching MF in multiple sets. The aim was to observe how differences in the intensity and proximity to MF would affect fatigue and strength recovery in men compared to women. It was hypothesized that women would return to the baseline quicker than men following a protocol without MF. However, when the proximity to MF was matched, it was hypothesized that no sex differences would occur.

## 2. Materials and Methods

### 2.1. Participants

A priori sample size calculation (*α* = 0.05, *β* = 0.8) determined a minimal sample size of 10 participants per group to detect a Cohen’s d of 1.2 in mean concentric velocity (MV) [17]. Twenty-two participants were enrolled at the start of the study, and a further three before the second part of data collection (10 women and 15 men). One man dropped out after baseline testing and a further eight men and five women were unable to complete the second part of data collection, leaving eight women and eight men who completed the submaximal RT-protocol (SMRT) and seven women and eight men who completed the RM RT-protocol (RMRT) (Table 1). Allocation to either SMRT or RMRT was randomized (random.org), stratified by sex. All participants were required to have at least six months of RT experience at a minimum frequency of three times per week and were able to squat at least bodyweight at baseline testing. Participants had to be free of any existing or residual lower body musculoskeletal injury for three months prior to testing.

### 2.2. Experimental Procedures

The study was designed based on a cross-over design requiring the participants to report to the facility 10 times over a period of six weeks following familiarization. During week one, baseline testing, consisting of body composition and back squat performance assessments (1RM and load-velocity profiling [18]), took place. Then, 48–72 h later, half of the participants (equal distribution of men and women) performed SMRT, which consisted of five sets of five repetitions (reps) at 80% of 1RM. The other half completed an RMRT, which consisted of five sets at a 4–6RM load to MF. Immediately before, 5 min, 24 h, 48 h, and 72 h after training, mean velocity (MV) in the BSq was measured at 80% of 1RM to assess recovery of neuromuscular function [19]. After a three week wash out period, it was intended that the groups would then perform the reverse RT protocol and post-RT testing sessions. However, due to the COVID-19 outbreak, data collection was suspended before the second RT-protocol could be performed and was postponed for six months. As a result, some participants were not able to participate in the second part of the data collection; those who did were required to perform a new baseline testing session, and more participants were recruited (Figure 1). Participants were required to refrain from any strenuous training 48 h before all testing. 

### 2.3. Baseline Testing

#### 2.3.1. Body Composition

Body composition was measured using an InBody 570 device (InBody, Cerritos, CA, USA), which has been found to be highly correlated with DXA readings (*r* = 0.93–0.98) [20]. Measurements were performed in accordance with the manufacturer’s instructions [20], with body mass (BM), fat mass (FM), and fat-free mass (FFM) being of primary interest. Near perfect test–retest reliability was observed (Intra Class Correlation (ICC) = 0.998–1.000).

#### 2.3.2. Load-Velocity Profiling and 1RM Strength Assessment

Following a standardized warm-up (Table 2), a progressive loading test up to the 1RM load was performed in the BSq exercise in accordance with previous work [18]. In short, with the barbell resting across the upper back, each participant descended in a self-determined fluent motion until the crease of the hip was below the top of the knee, as visually assessed by the head researcher. From this position, participants were then asked to perform the concentric phases as fast as possible. Initial load was set at 20 kg and was progressively increased in 10 kg increments until an MV of <0.6 m/s was reached, corresponding to ≈85% of 1RM [21]. MV was measured using the PUSH band^™^ 2.0 (PUSH Inc., Toronto, Canada). Thereafter, to determine the 1RM with more precision, the load was individually adjusted with smaller increments (2.5–5 kg) until the lifter was unable to complete the lift. An extra attempt at a given weight was offered to each participant. For safety reasons, each squatter had two to three spotters during each set, all of whom were experienced lifters or trainers. Three reps were executed for light (≥1.1 m/s), two for medium (0.6–1.1 m/s 1RM), and only one for the heaviest loads (<0.6 m/s). Verbal encouragement and visual velocity feedback were provided to motivate participants to ensure maximal effort. Participants rested at least 5 min between sets at velocities of below 0.6 m/s. The fastest MV at each load was considered for analysis. All velocity measures in this study are the MVs of the concentric phase of the lift. All performance testing and training was done with a free-standing squat rack (ER equipment, Albertslund, Denmark), a 20 kg powerlifting bar, and calibrated weight plates (Eleiko, Halmstad, Sweden). Excellent test–retest reliability was observed for 1RM (ICC = 0.998) and MV (ICC = 0.905). 

### 2.4. Performance Assessment and RT Protocols

Forty-eight–seventy-two h after baseline testing, participants reported to the facility and began performing the standardized warm-up (Table 2). For neuromuscular performance testing pre-RT, the participants then performed three reps at 80% of 1RM with maximal intended concentric velocity, of which the fastest MV was used as a measurement of baseline neuromuscular function. After five minutes of rest, half of the participants performed the SMRT protocol while the other half performed the RMRT protocol. Following another five minutes of rest, the participants performed another three reps at 80% of their 1RM, which established their neuromuscular function five minutes after training. The participants reported to the facility the following three days at the same time of day as their RT-session for post-testing. Post-testing consisted of the standardized warmup followed by three reps at 80% of 1RM with maximal intended concentric velocity, of which the highest MV was used for analysis of neuromuscular function at 24 h, 48 h, and 72 h after training.

#### 2.4.1. SMRT

The SMRT protocol consisted of five sets of five reps at 80% of 1RM with at least five minutes between sets in the BSq. The objective of this protocol was to assess the differences between men and women when volume, relative to maximal strength, was standardized.

#### 2.4.2. RMRT

The RMRT protocol consisted of five sets to MF at an initial 4–6RM load with at least five minutes between sets. MF was defined as the inability of the participant to complete the lift and required the assistance of the spotters. Following the warmup and baseline testing, the bar was loaded to 85% of 1RM. From here, the participants were instructed to perform sets of seven reps. If successful, the load was increased by 2.5%. This was repeated until failure was reached within 4–6 reps. The participants then completed four additional sets to failure with the same load with at least five minutes between sets, irrespective of the reps going below four reps. The objective of this protocol was to assess the differences between men and women when an equal amount of sets were performed until MF. The tonnage performed prior to the first counted set was termed pre-tonnage.

### 2.5. Instrumentation

MV was measured by the PUSH band^™^ 2.0, a smartphone-based wearable device designed to track movement velocity during a variety of resistance exercises, which has been determined to be both valid and reliable in the BSq [22]. The PUSH band^™^ 2.0 was located on the barbell in accordance with the manufacturer’s instructions. The concentric MV output from each rep was sent via Bluetooth to the Apple iPad proprietary PUSH application. The details of the PUSH band^™^ 2.0 computations have been described in detail elsewhere [23].

### 2.6. Statistical Analysis

All statistical analysis was performed in IBM SPSS Statistics 26 (IBM Corporations, Somers, NY, USA) except sample size calculations. Sample size calculations were done a priori using G*Power v3.1 computer software with MANOVA: Repeated-measures, within-between interaction computations. As a result of the methodological changes from the COVID-19 outbreak, a post hoc power analysis was performed, which resulted in a 64% probability of detecting a significant result with a Cohen’s d of 1.2. The assumption of homogeneity by Levene’s test and normality by Shapiro–Wilk’s test were met for all training and recovery variables. Baseline characteristics of the participants were presented using descriptive statistics (mean values and SDs). Recovery kinetics were presented as a percentage relative to baseline (% ±SD). To test for differences in baseline characteristics and training variables, a mixed linear modelling (MLM) was conducted with protocol and sex as fixed factors, and individual participants as random factors. To determine the effect of time, protocol, and sex on MV recovery, an MLM was conducted with protocol, sex, and time as fixed factors, and individual participants as random factors. Where significance was observed between fixed factors, Bonferroni post hoc tests were used for pairwise comparisons. Significance level was set a priori at *p* < 0.05. Effect sizes (ES) within-sexes was calculated with the Cohen’s d (pre-post difference divided by the pre-SD) [24]. Due to a small sample size, between-sex ES was calculated using the Hedge’s g [25]. The interpretations are trivial (<0.20), small (0.20–0.50), moderate (0.50–0.80), and large (>0.80).

## 3. Results

### 3.1. Training Variables

The between-group, between-sex, within-group between-sex, and within-sex between-group differences in the training variables are illustrated in Table 3 and Table 4. Despite no differences in the total training tonnage performed (*p* = 0.101), significantly fewer total reps (*p* < 0.001) at higher intensities (*p* < 0.001) were performed during the RMRT compared to the SMRT. No sex differences in the training intensity, total reps performed, or last rep MV were observed during the SMRT. However, in the RMRT, the women performed fewer total reps (*p* = 0.016), trained with higher intensities (*p* = 0.013), completed slower last reps than men (*p* = 0.044), and performed their last reps at slower velocities than their MEV_1RM_ (*p* = 0.012), while the last rep MV of the men did not differ from their MEV_1RM_ (*p* = 0.119).

### 3.2. Recovery Kinetics

A significant main effect of group (F(1, 136.4) = 24.166, *p* < 0.001), sex (F(1, 18.9) = 7.161, *p* = 0.015), and time (F(4, 128.1) = 3.793, *p* = 0.006) was observed. Furthermore, significant interaction effects of group × sex (F(1, 136.4) = 10.399, *p* = 0.002) and group × time (F(4, 128.1) = 2.733, *p* = 0.032) were found. Post hoc Bonferroni corrections revealed after the SMRT that men were significantly more fatigued than women at 5 min (*p* < 0.001, ES = 1.62), 24 h (*p* = 0.005, ES = 1.01), and 48 h (*p* < 0.001, ES = 2.25), but not 72 h (*p* = 0.061) (Figure 2). Following the RMRT, the men were significantly below the baseline at 5 min after training (*p* = 0.025, ES = 1.08); however, no differences between the men and women were observed at any timepoint (Figure 3). Additionally, the women were significantly more fatigued following the RMRT compared to the SMRT at 5 min (*p* < 0.001, ES = 1.70), 24 h (*p* = 0.003, ES = 1.36), 48 h (*p* = 0.002, ES = 1.34), and 72 h (*p* = 0.001, ES = 1.82) following exercise. No other significant effects of group, sex, or time were observed for the recovery kinetics.

## 4. Discussion

The main findings of this study were that: (1) following a protocol in which the volume was matched relative to maximal strength, the men were significantly more fatigued than women; (2) however, when an equal amount of sets were performed to muscular failure, no sex differences in the recovery were observed. Together, these results could suggest that the training stress required to produce a substantial amount of fatigue is larger for women, but, if the stress is of sufficient magnitude, few sex differences are apparent. However, the women trained with higher intensities and were able to complete slower last reps during the RMRT than both the men and their own 1RM. Thus, it cannot be ruled out that the results following the SMRT might not be a result of physiological sex differences in fatigability but a difference in the two groups’ ability to express their true 1RM strength (which may or may not be a sex difference).

The current investigation observed that, when resistance trained women and men perform five sets to MF with a 4–6RM load in the back squat, no sex differences are evident. However, when performing the same number of repetitions at 80% of their 1RM, a large sex difference exists. This is supported by multiple studies that observed no sex differences in the days following strength training [13,14,15,16]. While Häkkinen (1994) and Judge and Burke (2010) both observed within-sex time-effects that suggest the delayed recovery of strength performance in men in the days following exercise, they erroneously based their conclusions on within-sex time effects and not between-sex differences [26]. In support of programming variables having an impact on sex differences, Hakkinen (1994) did observe men to be more fatigued than women immediately following 10 sets of 10RM in the back squat. This effect was not seen following 20 sets of 1RM [14], suggesting a mediating role of volume and/or intensity on the observed sex differences in fatigability. In contrast, Davies et al. (2018) observed that women were more fatigued immediately and in the days after six sets, the last of which to MF, of the BSq at 80% of the 1RM. Excluding the results of Marshall et al. (2020) and Judge and Burke (2010) due to different training protocols and muscles tested, the results of the current study partly support those of the current literature. The combined results of the studies using squats as the fatiguing exercise might suggest that women are less impacted by training than men when the absolute and/or relative intensities are lower. However, when the intensity and/or fatigue is sufficiently high, any sex differences appear to be minimized. One explanation for the different results between Davies et al. (2018) and the remaining studies could be the sixth set to MF as women may be able to perform more repetitions at submaximal intensities [8,10]. Furthermore, going to MF at higher reps has been shown to produce more fatigue than lower reps [17]; however, as these data were not provided, this remains a hypothesis.

Another explanation could be differences in training status. Multiple physiological sex differences have been suggested to explain possible sex differences in fatigability and recovery; however, many of these are subject to change following exercise. Women have been shown to present with a higher type-I fiber area, higher concentrations of circulating estrogens, higher muscle perfusion, a lower reliance on the anaerobic metabolism, and be less sensitive to central fatigue [9,11,12]. While many of these factors have indeed been associated with fatigability and/or recovery [9,27,28], they or their association with recovery and/or fatigability are subject to change dependent on the contraction type, training intensity, training status, hormonal status, etc. [9,29,30,31]. Thus, the specifics of training history might either diminish or magnify the physiological differences observed between men and women.

The research on the sex differences in recovery following resistance training is equivocal as the studies favoring women are also those with the largest differences in the relative strength between the sexes. Thus, whether the observed differences are due to differences in biological sex or training status is unknown. As a result of the stigmatization of strength training for women, the participation in strength sports and the execution thereof in terms of volume, intensity, and effort may differ between the sexes [32,33,34], leaving training history as an unconsidered variable. This distinction is important as stronger individuals have been observed to be more fatigable [9], and, if individuals possess different abilities at expressing their maximal strength, it will lead to training at differing relative intensities when training is expressed relative to the maximum [11]. It is, therefore, important to discuss possible mechanisms in relation to both differences in sex-specific physiology and how training affects them. When looking at the research on sex differences in recovery kinetics, the studies suggesting a female advantage compared women with men who were, relative to the BM or FFM, 54–68% stronger than them [13,14,15], while no sex differences or a male fatigability advantage were apparent when the relative strengths were within 5–7% of each other [3,16].

When looking at the current study, it is apparent that the sexes were not equally strong. It has been suggested that women may be able to complete more reps to failure at submaximal intensities than men, especially below 80% of the 1RM [8,10]. Although not directly investigated, this hypothesis is partly supported by the training data in the current study. First, the women needed to use higher intensities (% of 1RM) during the RMRT to reach failure between four and six reps (89.7 vs. 87.4%), with all the men reaching a 4–6RM load between 85 and 90% of the 1RM, while women ranged between 87 and 95% of the 1RM, which is a difference that may have been even higher had the protocol not required some of the participants to perform multiple sets of seven reps beforehand. Second, when calculating the expected reps to failure for both the men and women from the velocity data obtained from both the SMRT and RMRT and adjusting for within-sex between-group differences in the MV at the 1RM and 80% at the baseline, it is approximated that the women were expected to be able to perform ≈4 reps more than the men at 80% of the 1RM (see equation), which may drastically alter the amount of fatigue experienced following the RT [17].
$$\mathrm{Expected}\;\mathrm{RM}=\frac{{\mathrm{Peak}\;\mathrm{MV}}_{\mathrm{SMRT}}-\left({\mathrm{Last}\;\mathrm{MV}}_{\mathrm{RMRT}}-\left({\mathrm{Group}\;\mathrm{difference}\;\mathrm{in}\;\mathrm{MEV}}_{1\mathrm{RM}}{\;\mathrm{and}\;\mathrm{MV}}_{80\%\;\mathrm{at}\;\mathrm{baseline}}\;\right)\right)}{\left(\mathrm{Average}\;\mathrm{velocity}\;\mathrm{loss}\;\mathrm{per}\;\mathrm{rep}\right)}$$

This difference between the sexes may be a result of biological differences; however, differences in training status cannot be ruled out. First, the men were relatively stronger than the women in this study (2.02 vs. 1.68 1RM BSq (kg)/FFM (kg)). Second, the men in the RMRT group produced at similar MVs at both their 1RM and their last rep before failure, while the women were able to complete a rep at velocities of 0.08–0.1 m/s slower than their 1RM MV. The ability to grind through a heavy rep is typically seen in more skilled lifters [35], possibly due to technical or psychological factors. This suggests that the women in this sample may not be habituated to near-failure lifting at maximal intensities. Thus, it is likely that the women were subjected to a lower relative intensity than the males during the SMRT, at least partly explaining the large sex differences observed.

When studying women, multiple methodological measures have been suggested to be implemented to decrease the risk of different maturation, menstrual cycle, and contraceptive statuses impacting the observed results, thereby increasing the internal validity of the studies [36,37]. While these factors may impact the individual responses to training, on average, the effects of the menstrual cycle phase and oral contraceptive status on variations in performance and fatigability are trivial or small at best [38,39,40,41]. The impact on recovery, however, is less studied. Only a few studies to date, using hormonal verification, have compared the recovery of neuromuscular performance following strength exercise across the menstrual cycle and oral contraceptive phases [42,43,44,45]. In untrained women following eccentric exercise, some research has found a decreased recoverability in the early follicular phase compared to the ovulatory phase [42], while other studies observe no differences between the early follicular phase and the ovulatory phase or midluteal [43,44]. The same is observed with oral contraception, where some [44], but not all [43], find retarded recovery in untrained women on oral contraceptives. Furthermore, no effect of the oral contraceptive phase, in trained women, has been seen on the recovery of neuromuscular function following a high volume squat protocol [45]. When measuring the blood markers of muscle damage in untrained women, more muscle damage may occur in the early follicular phase [42,43,46]; however, this has not been observed in trained women [47]. Thus, while some studies suggest decreased recoverability in the early follicular phase and with oral contraceptives, others fail to observe such a difference. Interestingly, it has been observed that 8 weeks of strength training in previously untrained women led to smaller fluctuations in estrogen and progesterone across the menstrual cycle [48]. This might explain why some studies in untrained women observe an effect of the menstrual cycle phase or contraceptive status, while no differences are observed in those using trained participants. It cannot be ruled out that hormonal status may have impacted the results of the current study as no effort was made to correct for this. Future studies should try to investigate how the menstrual cycle phase and oral contraceptive status affect recovery kinetics, preferably in trained women for whom these details matter most as hormonal status, fatigability, and recovery may all be impacted by training status [9,48,49].

This study is not without its limitations. First, due to the COVID-19 lockdown, the second part of data collection was postponed, resulting in large drop-out rates, decreasing the statistical power and possibly altering the training status of the remaining participants. In an attempt to mitigate this [50], modifications were made: (1) although not enough to satisfy a priori power calculations, additional subjects were recruited. When observing the effect sizes, it seems that the statistical power was sufficient at detecting the between-sex magnitudes as they were large following the SMRT and trivial following the RMRT; (2) the subjects participating in both data collections were required to undergo additional baseline testing as changes in training status might have occurred; and (3) to take into account that some individuals were included in both trials and others were not, the statistical analyses was changed from a repeated-measures MANOVA to a mixed linear model. Second, the sole focus of this study was to investigate the differences in fatigue, measured by lifting velocity, following different training protocols, which leaves mechanistic explanations for the observed results outside the scope of this article. To explain the observed results, many measures could be of interest. First, the back squat is a relatively complex movement requiring participants to balance with weight; therefore, force plate data examining weight distribution on their feet as a measure of technical abilities could be of interest as lower technical abilities may prevent individuals from experiencing true MF in complex movements and, therefore, accumulating less fatigue. Second, measures of central and peripheral fatigue, e.g., the twitch interpolation technique, measures for substrate depletion, muscle oxygenation, etc., would allow for explaining how and if the sexes fatigue through different mechanisms. Third, it would be interesting to see if any associations between the fiber type composition, blood flow, capillary density, mechanical arterial compression, substrate depletion, and recovery exist and especially whether and/or how these correlations would change with training status.

## 5. Practical Applications

In practice, the current results may have two primary takeaways. One, if using RM-zone based training, which requires the athlete to perform all or multiple sets to muscular failure at higher intensities, no sex differences in programming seem necessary with regards to recovery. Two, if using percentage-based or velocity-based programming, coaches should recognize that different individual athletes and sexes may be able to perform a different number of reps and have different maximal effort velocities at different intensities. Therefore, in addition to 1RM testing and/or the creation of individual load-velocity profiling, repetition maximum testing at lower intensities may be needed to gauge the proximity to failure and gather velocity data before and/or during a training program to ensure the desired intensity of training is performed.

## 6. Conclusions

In conclusion, following a submaximal protocol in which the volume was equal relative to the maximal strength, men experienced more fatigue than women. However, when exposed to five sets to muscular failure, no sex differences in fatigue or recovery were observed. Caution must be taken when interpreting these results as the relative strength was not matched and the results cannot be reduced to pure biological sex differences. Future research should focus on the impact of training history on the magnitudes of sex differences in response to exercise, whether specific training can alter this response, and if sex differences in load-velocity and fatigue-velocity profiles exist in equally trained populations. Furthermore, the inclusion of mechanistic measures could help explain any possible sex differences.

## Figures and Tables

**Figure 1 sports-09-00157-f001:**
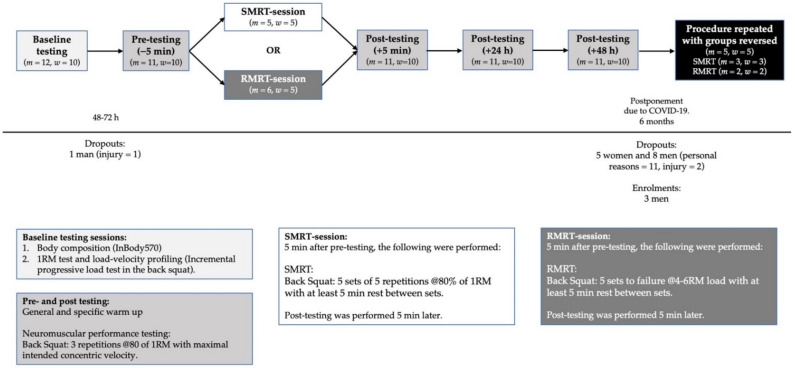
Graphical abstract of study proceedings.

**Figure 2 sports-09-00157-f002:**
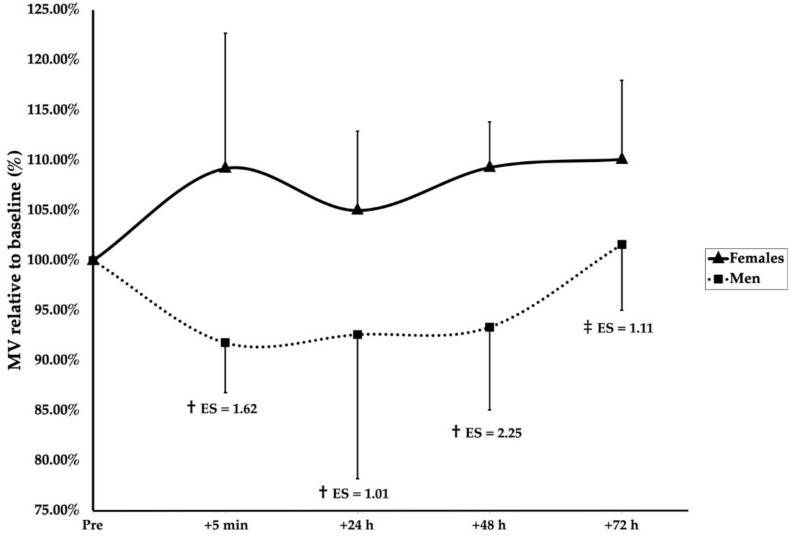
Recovery kinetics following SMRT. † indicates a significant sex difference (*p* < 0.05). ‡ indicates a tendency for sex differences (*p* = 0.061). Effect sizes (ES) are between-sex magnitudes.

**Figure 3 sports-09-00157-f003:**
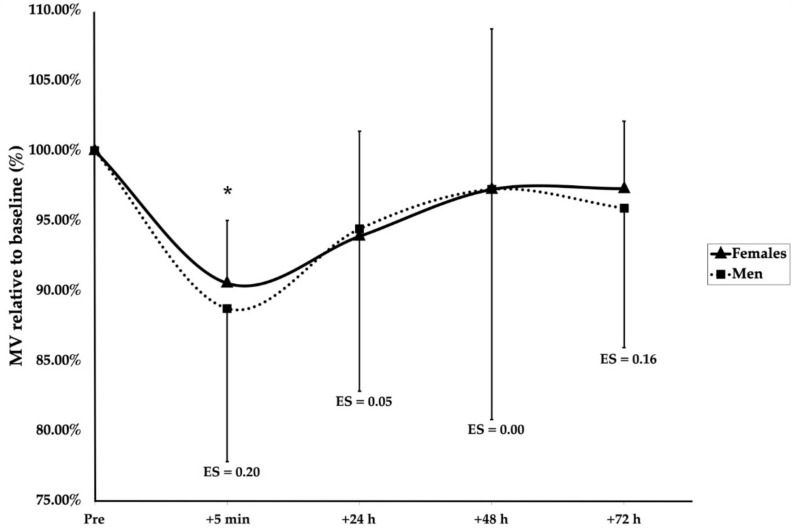
Recovery kinetics following RMRT. * indicates a significant time effect for males (*p* < 0.05). ES are between-sex magnitudes.

**Table 1 sports-09-00157-t001:** Participant characteristics. Data are presented as mean ± SD.

	SMRT Group (*n =* 16)		RMRT Group (*n =* 15)	
	Men (*n* = 8)	Women (*n* = 8)	Men (*n* = 8)	Women (*n* = 7)
Age (years)	29 ± 5	26 ± 4	28 ± 4	25 ± 4
Height (cm) *	182 ± 9 †	165 ± 9 †	182 ± 5 †	161 ± 8 †
Body Mass (kg) *	86 ± 14.4 †	70.3 ± 8.8 †	82.8 ± 6.2 †	61.8 ± 6 †
Fat Mass (kg)	14.3 ± 3.3	18.7 ± 8.2	11.2 ± 4.6	14.3 ± 5.2
Fat Free Mass (kg) *	71.7 ± 13 †	51.6 ± 7.6 †	71.6 ± 6.1 †	47.4 ± 5.7 †
1RM Back Squat (kg) * §	141.6 ± 29.0 †	85.1 ± 18.8 †	148.1 ± 26.5 †	80 ± 8.4 †
Relative strength (kg/kg) ^ *	1.97 ± 0.16 †	1.67 ± 0.35 †	2.07 ± 0.32 †	1.7 ± 0.24 †
MEV_1RM_ (m/s) ^^	0.33 ± 0.07	0.34 ± 0.09	0.37 ± 0.08	0.41 ± 0.1
Baseline MV_80% of 1RM_ (m/s) *	0.72 ± 0.09 †	0.58 ± 0.1 †	0.71 ± 0.08	0.67 ± 0.1

^ Relative strength = 1RM Back Squat/Fat Free Mass. ^^ MEV_1RM_ = The mean concentric velocity of their 1RM lift. * indicates a significant overall difference between the sexes (*p* < 0.05). § indicates a significant overall difference between the groups (*p* < 0.05). † indicates a significant between-sex within-group difference (*p* < 0.05).

**Table 2 sports-09-00157-t002:** Overview of the standardized warm-up protocols.

Standardized Warm-Up Routines	
General warm-up	2 rounds of 5 reps of: Cossack Squat (per side), Kang Squat, Inch Worms, World’s Greatest Stretch (per side), and Off-set Squat with a 5 kg plate in front.
Specific warm-up	3 reps in the BSq with maximally intended concentric velocity with: 20 kg bar, 45%, 50%, 60%, and 70% of 1RM

**Table 3 sports-09-00157-t003:** Training variables. Data are presented as mean ± SD.

	SMRT Group (*n =* 16)		RMRT Group (*n =* 15)	
	Males (*n* = 8)	Females (*n* = 8)	Males (*n* = 8)	Females (*n* = 7)
Total tonnage performed (kg) *	2787.5 ± 272 †	1673.8 ± 371.5 †	2911.4 ± 389.6 †	1873 ± 181 †
Pre tonnage (kg) §	0 ± 0 €	0 ± 0 €	889.4 ± 462.6 €	893.4 ± 552.3 €
Training load (kg) *§	113.3 ± 23.2 † €	68.1 ± 15 † €	130.0 ± 23.5 † €	71.7 ± 7.4 † €
Training intensity (% of 1RM) §	80 ± 0 €	80 ± 0 €	87.7 ± 1.5 † €	89.7 ± 2.6 † €
Total reps completed § *	25 ± 0 €	24 ± 1.1 €	17.1 ± 1.6 † €	15.1 ± 2.3 † €
Mean Set Velocity Loss (%)	19.3 ± 7.2	21.0 ± 5.4	18.8 ± 6.1	15.9 ± 8.3
Mean Velocity Increase Between Sets (%)	25.6 ± 8.8	29.1 ± 11.2	23.2 ± 11.2	18.2 ± 18.4
Last MV_mean_ (m/s) § * Last MV_mean_–MEV_1RM_ (m/s) § *	0.53 ± 0.1 € 0.2 ± 0.08 €	0.46 ± 0.1 € 0.12 ± 0.09 €	0.38 ± 0.06 † € 0.02 ± 0.08 † €	0.32 ± 0.05 † € −0.1 ± 0.12 † €

* indicates a significant overall difference between the sexes (*p* < 0.05). § indicates a significant overall difference between the groups (*p* < 0.05). † indicates a significant sex difference within groups (*p* < 0.05). € indicates a significant group difference within sexes (*p* < 0.05).

**Table 4 sports-09-00157-t004:** Repetition and velocity performance across sets. Data are presented as mean ± SD.

	SMRT Group (*n* = 16)		RMRT Group (*n* = 15)	
	Males (*n* = 8)	Females (*n* = 8)	Males (*n* = 8)	Females (*n* = 7)
**Set 1**				
Reps performed §	5 ± 0	5 ± 0 €	4.8 ± 0.7 †	4.3 ± 0.5 † €
Peak MV (m/s) § *	0.66 ± 0.04 €	0.60 ± 0.14 €	0.51 ± 0.09 † €	0.41 ± 0.06 † €
Last MV (m/s) §	0.52 ± 0.12 €	0.46 ± 0.13 €	0.38 ± 0.09 €	0.32 ± 0.05 €
**Set 2**				
Reps performed § *	5 ± 0 €	5 ± 0 €	4.3 ± 0.5 † €	3.3 ± 0.8 † €
Peak MV (m/s) § *	0.66 ± 0.09 † €	0.57 ± 0.09 † €	0.49 ± 0.06 † €	0.4 ± 0.04 † €
Last MV (m/s) § *	0.53 ± 0.09 † €	0.44 ± 0.1 † €	0.40 ± 0.08 €	0.33 ± 0.06 €
**Set 3**				
Reps performed §	5 ± 0 €	5 ± 0 €	3.5 ± 0.8 €	3 ± 1.2 €
Peak MV (m/s) § *	0.66 ± 0.09 €	0.59 ± 0.16 €	0.49 ± 0.1 † €	0.38 ± 0.09 † €
Last MV (m/s) §	0.52 ± 0.09 €	0.47 ± 0.11 €	0.4 ± 0.07 €	0.33 ± 0.07 €
**Set 4**				
Reps performed §	5 ± 0 €	4.9 ± 0.4 €	2.6 ± 0.9 €	2.6 ± 0.5 €
Peak MV (m/s) § *	0.66 ± 0.09 €	0.59 ± 0.16 €	0.46 ± 0.07 €	0.37 ± 0.08 €
Last MV (m/s) § *	0.55 ± 0.12 €	0.46 ± 0.11 €	0.36 ± 0.07 €	0.3 ± 0.05 €
**Set 5**				
Reps performed §	5 ± 0 €	4.8 ± 0.7 €	2 ± 0.9 €	2 ± 0.8 €
Peak MV (m/s) § *	0.65 ± 0.07 † €	0.57 ± 0.11 † €	0.43 ± 0.07 † €	0.34 ± 0.10 † €
Last MV (m/s) §	0.54 ± 0.12 €	0.47 ± 0.1 €	0.36 ± 0.09 €	0.30 ± 0.06 €

* indicates a significant overall difference between the sexes (*p* < 0.05). § indicates a significant overall difference between the groups (*p* < 0.05). † indicates a significant sex difference within groups (*p* < 0.05). € indicates a significant group difference within sexes (*p* < 0.05).

## Data Availability

The data presented in this study are available on request from the corresponding author.

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
