# Peer review of "Impact of Training Protocols on Lifting Velocity Recovery in Resistance Trained Males and Females"

_sports, 2021, doi:10.3390/sports9110157_

Round 1
Reviewer 1 Report
The work is devoted to the problem of the training process of men and women. The problem considered in the manuscript is extremely urgent.
However, it has a number of serious disadvantages.
The presented manuscript has a big drawback, since no material is presented that reflects changes in electromyographic activity when performing various motor tasks. Accordingly, the mechanism that determines these changes is not presented. Recording electrical activity is simply an integral part of such research.
Give a transcript of abbreviations.
And the most important thing in the work is completely ignored such an important factor as the hormonal factor associated with the menstrual cycle.
Reviewer 2 Report
The work is interesting and can be of practical importance. It requires editorial corrections for good reception.
- The experiment procedure is described in great detail. Supported by a drawing. HOWEVER, the DESCRIPTION itself is very illegible. The description should be organized. MAYBE a SURE solution would be to include a clear description containing: exercise, load, number of repetitions, rest break time in the table. Then, instead of a long description, we will receive information allowing us to visualize this experiment. The warm-up is well described. Research organization should not be confused with the organization - the content of the experimental procedures.
- It is difficult to understand why the paper presents information related to the disruption of the experiment due to the COVID19 pandemic. This has no effect on the results. This would be justified if the results of the group were compared before and after Covid19. I believe that this content should be removed from the work. They do not provide any information for the problem being solved. 3. Comparative work on strength training in women and men cannot ignore the hormonal status of the respondents. For this reason, I believe that the omission of this issue limits the possibilities of inference. Putting this thread of work in the paragraph on constraints is insufficient.
- The attempt to assess the impact of the COVID19 pandemic on research results is incomplete. We do not know what the pre-pandemic and pandemic training looked like until the start of the research. There is no information about the possible infection of the respondents with COVID19.
- Restrictions should be separated into a separate part of the work.
- In a very extensive discussion, the authors focus on comparing the results obtained in the group of women and men. They try to justify the differences not unequivocally. A constant element of doubts is the reliability of the load selection in terms of its equivalence in both sexes.
- The lack of research in the assessment of the actual load on the energy system, circulation, respiratory system, the picture of blood flow through the muscles is an interpretation barrier. The work shows the multidirectional measurements that must be carried out in this type of comparative research.
- A good practical conclusion would be to identify all the measurements that were missing. Measurements that would allow a scientific explanation of the observed phenomena. I believe that the work requires a large editorial correction in the direction of:
- simplification of the description of the research organization and the methodology of the experiment
- exclusion from the content of the COVID19 issue as being not work related
- separation of the limitation part - literature analysis regarding hormonal differences between men and women in the context of strength training
- separation of practical recommendations for the scope of comparative studies between men and women in research on strength training
Round 2
Reviewer 2 Report
THANK YOU VERY MUCH FOR YOUR ANSWER. I understand and respect the position of the authors in the context of the submitted comments. I believe that the text meets the criteria for publication in the Sport journal